# Structure of BAI1/ELMO2 complex reveals an action mechanism of adhesion GPCRs via ELMO family scaffolds

Zhuangfeng Weng[1,2], Chenghao Situ[3], Lin Lin[1], Zhenguo Wu[3], Jinwei Zhu[1] & Rongguang Zhang[1,2]

The brain-specific angiogenesis inhibitor (BAI) subfamily of adhesion G protein-coupled receptors (aGPCRs) plays crucial roles in diverse cellular processes including phagocytosis, myoblast fusion, and synaptic development through the ELMO/DOCK/Rac signaling pathway, although the underlying molecular mechanism is not well understood. Here, we demonstrate that an evolutionarily conserved fragment located in the C-terminal cytoplasmic tail of BAI-aGPCRs is specifically recognized by the RBD-ARR-ELMO (RAE) supramodule of the ELMO family scaffolds. The crystal structures of ELMO2-RAE and its complex with BAI1 uncover the molecular basis of BAI/ELMO interactions. Based on the complex structure we identify aGPCR-GPR128 as another upstream receptor for the ELMO family scaffolds, most likely with a recognition mode similar to that of BAI/ELMO interactions. Finally, we map disease-causing mutations of BAI and ELMO and analyze their effects on complex formation.

[1] State Key Laboratory of Molecular Biology, CAS Center for Excellence in Molecular Cell Science, Shanghai Institute of Biochemistry and Cell Biology, Chinese Academy of Sciences; Shanghai Science Research Center, University of Chinese Academy of Sciences, 333 Haike Road, 201210 Shanghai, China. [2] School of Life Science and Technology, ShanghaiTech University, 100 Haike Road, 201210 Shanghai, China. [3] Division of Life Science, Center for Stem Cell Research, Center for Systems Biology and Human Health, the State Key Laboratory in Neuroscience, Hong Kong University of Science and Technology, Kowloon, Hong Kong, China. Correspondence and requests for materials should be addressed to J.Z. (email: jinwei.zhu@sibcb.ac.cn) or to R.Z. (email: rgzhang@sibcb.ac.cn)

Adhesion G protein-coupled receptors (aGPCRs), which include 33 members in human genome, are the second-largest group within the superfamily of GPCRs[1]. The most prominent signature feature of aGPCRs is that, in addition to the typical seven-transmembrane helical architecture, they all contain a large N-terminal extracellular region with multiple structural domains that are required for mediating cell-cell or cell-matrix interactions[1,2]. The extended N-terminus can be cleaved at the GPCR proteolysis site (GPS) within the GPCR autoproteolysis-inducing (GAIN) domain, thus facilitating a variety of intracellular signaling transductions[3]. aGPCRs are widely expressed and play critical roles in cell adhesion, cell polarity and cell migration[2,4,5]. Abnormalities of aGPCRs have been linked with various human diseases including cancers[6,7], immunological diseases[8], nephrotic syndrome[9], and psychiatric disorders[10], etc.

The brain-specific angiogenesis inhibitors 1-3 (BAI1-3, also known as ADGRB1-3) comprise one of the most extensively studied subfamilies of aGPCRs[11]. BAI1 was initially identified as a target gene of the tumor suppressor p53, an inhibitor of angiogenesis, and a growth suppressor of glioblastomas[12,13]. Later, BAI1 was discovered as an engulfment receptor that can specifically recognize phosphatidylserine, a key eat-me signal exposed on apoptotic cells, and promote the internalization of apoptotic cells through the ELMO/DOCK/Rac signaling module[14]. BAI1 was also identified as a pattern recognition receptor for phagocytosis of gram-negative bacteria by macrophages in response to pathogen infection[15]. Similarly, the bacterial uptake is also triggered by BAI1-mediated activation of Rac signaling in an ELMO/DOCK-dependent manner[15]. Recently, two lines of unexpected discoveries reported that both of BAI1 and BAI3 can promote myoblast fusion by means of the ELMO/DOCK/Rac signaling, suggesting a crucial role of BAIs in muscle development and repair[16,17]. Moreover, all of the BAI subfamily members are enriched in the post synaptic density (PSD), a central hub for neuronal signaling transduction in the excitatory synapses[18]. Mice lacking of BAI1 led to a thinning of PSD in hippocampal neurons and deficits in spatial learning and memory[19]. Loss of BAI3 resulted in a deficit in dendritic arbor formation and synapse maturation[20]. Notably, BAIs play essential roles in synaptogenesis and synaptic plasticity at least in part via their interactions with the ELMO scaffolds[20,21].

The above discoveries implied that the ELMO/DOCK/Rac signaling module might serve as a common pathway downstream of BAIs, independent of G protein signaling. The ELMO family scaffolds contain three members, ELMO1-3. Each member consists of the Ras-binding domain (RBD), the armadillo repeats domain (ARR), the engulfment and motility (ELMO) domain, the pleckstrin homology (PH) domain and the C-terminal proline rich region (PRR)[22]. It is well known that binding of PH domain to SH3 domain of DOCKs, a family of evolutionarily conserved atypical Rho guanine nucleotide exchange factors (GEFs) for Rac and/or Cdc42[23], would relieve the autoinhibited conformation of DOCKs, thus facilitating the activation of Rac signaling[24]. Rac-dependent actin cytoskeletal remodeling is believed to be pivotal for the aforementioned processes like phagocytosis, myoblast fusion, dendritic spine remodeling, cell migration, etc[11]. However, the molecular basis of how ELMO/DOCK/Rac axis couples to BAI subfamily aGPCRs remains elusive. Perhaps a more interesting question is whether this signaling module is also employed by other aGPCR(s), if so, whether the underlying molecular mechanism is the same as those of BAI/ELMO assemblies.

In the present study, we perform a detailed biochemical characterization of the BAI1/ELMO2 interaction, and find that an evolutionarily conserved fragment located in the intracellular region of BAI1 specifically binds to the RBD-ARR-ELMO (RAE) tandem of ELMO2. To elucidate the assembly of BAI1/ELMO2 complex, we solve the crystal structures of the RAE tandem of ELMO2 and its complex with BAI1 at 2.5-Å and 1.7-Å resolutions, respectively. The RBD, ARR and ELMO domains of ELMO2 bind tightly and form a structural and functional supramodule, creating a highly conserved elongated concave groove capable of specifically binding to the BAI1 fragment. The structures determined here reveal a previous uncharacterized interaction mode for the armadillo repeats, and also establish a framework for dissecting the function and disease mechanism of the two family proteins. Moreover, based on the structure of BAI1/ELMO2 complex, we predict and verify that another aGPCR-GPR128 (also known as ADGRG7) also binds to the RAE supramodule of ELMO family scaffolds using a similar mode, suggesting that recruitment of the ELMO/DOCK/Rac signaling module is likely a common feature of some aGPCRs subfamilies.

## Results

**ELMO2 specifically interacts with an intracellular fragment of BAI1.** A cytosolic fragment of BAI1 (aa 1431-1582, Fig. 1a) was reported to bind to the N-terminal fragment (aa 1-558) of ELMO1 which includes the RBD, ARR and ELMO domains[14] (Fig. 1b). The interaction is conserved among isoforms of BAIs and ELMOs[17]. We first sought to verify the interaction. We confirmed the binding by showing that GST-tagged BAI1[1431-1582] bound robustly to Flag-tagged RBD-ARR-ELMO tandem of ELMO2 (aa 1-520, referred to as ELMO2-RAE) (Fig. 1c). However, further deletion of the ELMO domain totally eliminated the interaction. Moreover, neither the ARR domain or the ELMO domain alone was able to bind to BAI1 (Fig. 1c). Taken together, these data indicated that the RBD, ARR, ELMO domains of ELMO2 form a structural and functional supramodule required for BAI1 binding.

We next sought to map the minimal fragment of BAI1 for ELMO2 binding. Based on a truncation-based approach, a 36-residue fragment of BAI1 (aa 1467-1502; referred to as the ELMO-binding domain (EBD), Fig. 1a) was sufficient to bind to the ELMO2-RAE (Fig. 1d). Further deletion of BAI1-EBD at its C-terminus (aa 1467-1491) diminished the binding (Fig. 1d). In line with the previous report, the EBD domains of BAI1-3 all bound to ELMO2-RAE effectively (Fig. 1e). We used the isothermal titration calorimetry (ITC)-based assay to measure the binding affinity between the two proteins, and the result showed that BAI1-EBD binds to ELMO2-RAE with a dissociation constant ($K_d$) of ~3.36 μM (Fig. 1f). In addition, the ITC data also indicated that BAI1-EBD binds to ELMO2-RAE in a 1:1 molar ratio.

**Structure of the ELMO2-RAE supramodule.** To obtain structural insights into supramodular formation of ELMO2-RAE, we first sought to crystallize it. Diffraction-quality crystals of ELMO-RAE belong to the $C222_1$ space group (Table 1). The structure was determined by single-wavelength dispersion (SAD) using selenomethione-substituted protein crystals, and then refined with the native data to 2.5-Å resolution (Table 1).

In the structure, the ELMO2-RAE forms an elongated super-helical architecture that consists of three domains: the N-terminal RBD domain (aa 1-80), the ARR domain (aa 81-301), and the ELMO domain (aa 306-513) (Fig. 2a, b). The RBD domain adopts a typical ubiquitin-like superfold (Fig. 2a and Supplementary Fig. 1a), which closely resembles the RBD of FHOD1 (Supplementary Fig. 1b), consistent with the previous bioinformatic analyses[22]. RhoG was reported to recruit ELMO/DOCK to activate Rac signaling via binding to N-terminal portion of

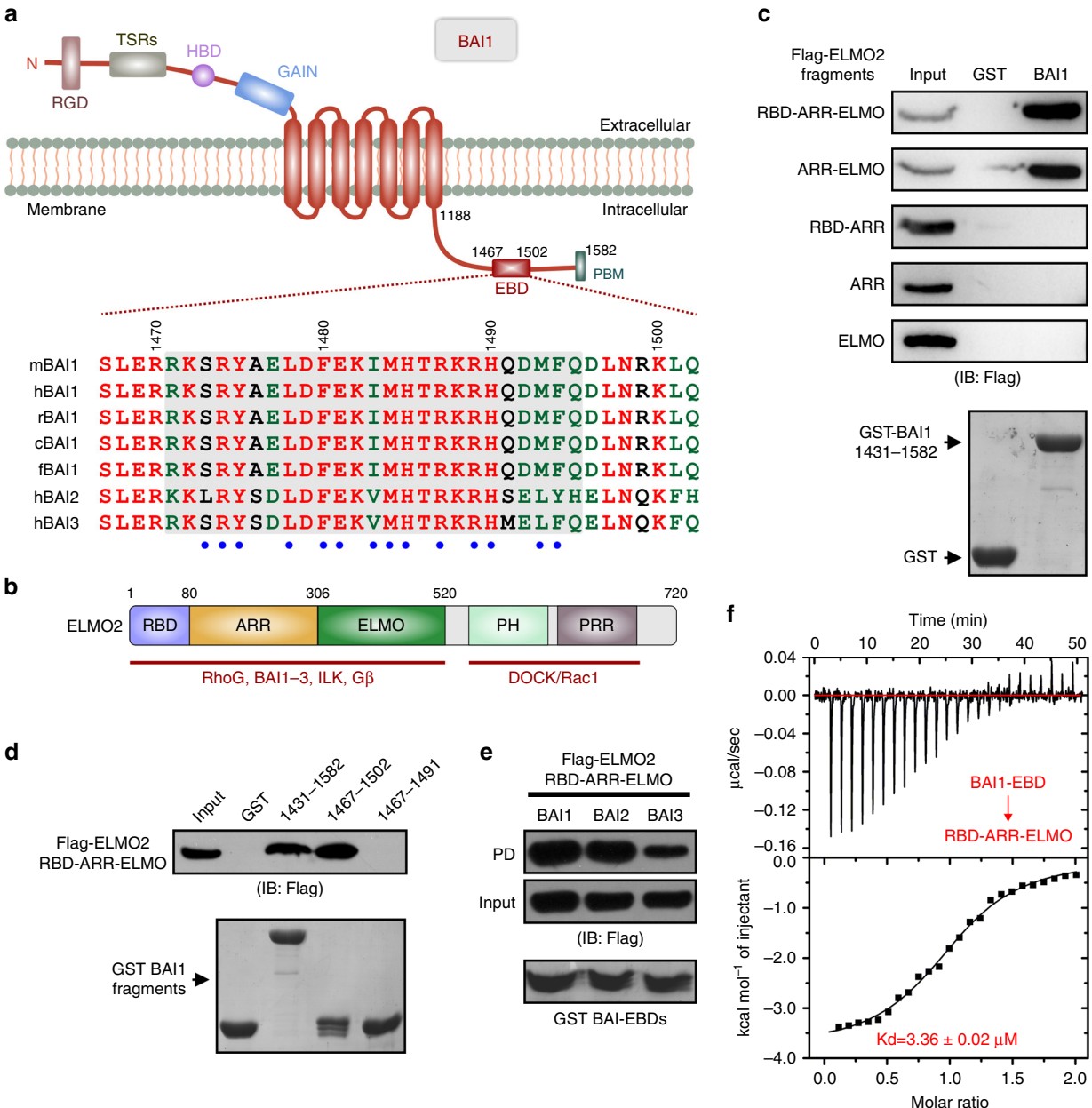

**Fig. 1** An intracellular fragment of BAI1 is specifically recognized by ELMO2. **a** Domain organization of BAI1. RGD, Arg-Gly-Asp integrin-binding motif; TSR, thrombospondin type 1 repeat; HBD, hormone-binding domain; GAIN, GPCR autoproteolysis-inducing domain; EBD, ELMO-binding domain; PBM, PDZ domain-binding motif. In the drawing, the structure-based sequence alignment of BAI1-EBD from different species and orthologs is shown. The absolutely conserved and conserved residues are colored red and green, respectively. Residues involved in binding to ELMO2 are annotated below as blue dots. The residues with clear electron densities in the structure are highlighted with gray box. **b** Domain organization of ELMO2. Notably, the N-terminus binds to RhoG, BAIs, ILK, and Gβ; The C-terminus binds to the DOCK/Rac axis. **c**, GST-pull down assays showing that the intact RBD-ARR-ELMO (RAE) tandem of ELMO2 is required for BAI1 recognition. **d** GST-pull down assays showing that the fragment aa 1467-1502 of BAI1 is the minimal region responsible for ELMO2-RAE binding. **e** All the BAIs (BAI1, 2, 3) bind to ELMO2-RAE effectively in the GST-pull down assays. PD, pull down. **f** Quantitative measurement of the binding affinity between BAI1-EBD and ELMO2-RAE using the ITC-based assay

ELMO1 (aa 1-362, including the intact RBD and ARR domain and partial ELMO domain)[25]. Given the typical Ras-like GTPase-binding site on ELMO2-RBD is remote from the ARR and ELMO domains (Fig. 2a and Supplementary Fig. 1b), one would expect that ELMO2-RBD is sufficient for RhoG binding. Indeed, we verified that ELMO-RBD bound to active RhoG as effectively as ELMO-RAE did (Supplementary Fig. 1c). The ARR domain contains five armadillo repeats (ARMs), though the ARM-1 is not complete (Supplementary Fig. 2). Superposition of ELMO2-ARR with FHOD1-ARR reveals the two structures are quite similar

despite low sequence homology (Supplementary Fig. 2c). The ELMO domain has an all α-helical topology, composed of 11 α-helices (Fig. 2c). A Dali[26] search suggests that the ELMO domain represent a novel fold with a global architecture. Notably, a 16-residue flexible loop ($^{421}$LQVGELPNEGRNDYHP$^{436}$) between α6 and α7 of the ELMO domain interacts tightly with the ARR domain (see below for the detail) (Fig. 2c).

**Binding interfaces between ELMO2-RAE supramodule.** In line with our biochemical data, the RBD, ARR and ELMO domains

**Table 1 Data collection and refinement statistics**

**Data collection**

| Dataset | ELMO2-RAE (SeMet) | ELMO2-RAE (Native) | ELMO2/BAI1 |
|---|---|---|---|
| Space group | C222$_1$ | C222$_1$ | P3$_1$21 |
| Unit cell (a,b,c,Å) | 75.818,118.568,170.027 | 75.182, 119.134, 169.326 | 92.869, 92.869, 130.762 |
| Unit cell (α,β,γ,°) | 90, 90, 90 | 90, 90, 90 | 90, 90, 120 |
| Wavelength (Å) | 0.97853 | 0.97853 | 0.97853 |
| Resolution range (Å) | 50.00–3.05 (3.10–3.05) | 50.00–2.48 (2.52–2.48) | 50.00–1.70 (1.73–1.70) |
| No. of unique reflections | 14767 (723) | 27341 (1366) | 72368 (3592) |
| Redundancy | 6.5 (6.4) | 12.6 (12.4) | 9.9 (10.0) |
| I/σ | 19.6 (2.2) | 28.0 (2.2) | 28.4 (2.9) |
| Completeness (%) | 100.0 (100.0) | 99.9 (99.8) | 100.0 (100.0) |
| Rmerge (%) [a] | 12.0 (64.5) | 6.8 (92.0) | 6.8 (64.0) |

**Structure refinement**

| | | | |
|---|---|---|---|
| Resolution, Å | | 42.33–2.48 (2.57–2.48) | 46.43–1.70 (1.76–1.70) |
| R$_{work}$[b]/R$_{free}$[c](%) | | 20.57 (28.97)/23.97 (30.47) | 18.28 (24.28)/20.80 (26.60) |
| Rmsd bonds/angles (Å/°) | | 0.004/0.733 | 0.011/1.27 |
| Average B factor (Å$^2$) | | 71.1 | 27.5 |
| No. of protein atoms | | 3996 | 4189 |
| No. of solvent molecules | | 26 | 397 |
| Favored regions | | 97.6 | 98.3 |
| Allowed regions | | 2.4 | 1.7 |
| Outliers | | 0 | 0 |

Numbers in parentheses represent the value for the highest resolution shell
[a]R$_{merge}$ = $\sum|Ii - Im|/\sum Ii$, where Ii is the intensity of the measured reflection and Im is the mean intensity of all symmetry related reflections
[b]R$_{cryst}$ = $\sum||Fobs| - |Fcalc||/\sum|Fobs|$, where Fobs and Fcalc are observed and calculated structure factors
[c]R$_{free}$ = $\sum T||Fobs| - |Fcalc||/\sum T|Fobs|$, where T is a test data set of about 5% of the total reflections randomly chosen and set aside prior to refinement

bind tightly in the structure (Fig. 3a). In the RBD-ARR binding interface, Y60$^{RBD}$ inserted into the hydrophobic pocket formed by Pro81, Ala84, Leu88, Leu110, and Phe116 from ARR domain, which is similar to RBD-ARR interface occurred in FHOD1[27] (Fig. 3b, c). In addition, the mainchain of Y60$^{RBD}$ forms a hydrogen bond with the sidechain of D113$^{ARR}$. A specific cation-π interaction was observed between Y48$^{RBD}$ and R83$^{ARR}$ (Fig. 3c). The ARR-ELMO binding interface could be separated into two parts: (1) a 16-residue flexible loop between α6 and α7 of ELMO domain forms extensive hydrogen bonds with the residues from α14 of ARR domain, such as H284$^{ARR}$-R431$^{ELMO}$ and N295$^{ARR}$-E425$^{ELMO}$ interactions (Fig. 3d). (2) the distal side of ARR domain binds tightly to ELMO domain via extensive hydrophobic interactions. For example, I267$^{ARR}$, I272$^{ARR}$, L286$^{ARR}$, together with L289$^{ARR}$ form hydrophobic interactions with Y434$^{ELMO}$ and F439$^{ELMO}$ (Fig. 3e). The second interface is further reinforced by additional polar interactions like R264$^{ARR}$-D442 $^{ELMO}$ and R300$^{ARR}$-D375$^{ELMO}$ pairs (Fig. 3e). It is worth noting that most of the residues involved in the binding interfaces are conserved among the ELMO family proteins (Supplementary Fig. 3), suggesting that the supramodular folding of RAE tandem is a common feature for all ELMOs.

**Crystal structure of the BAI1/ELMO2 complex**. To delineate how then the BAI1-EBD is recognized by this ELMO2-RAE supramodule, we solved the crystal structure of ELMO2-RAE in complex with a synthetic BAI1-EBD peptide at 1.7-Å resolution using the molecular replacement method (Table 1). Each asymmetric unit contains one BAI1/ELMO2 complex with 1:1 stoichiometry, consistent with our biochemical data (Fig. 1f). There is no significant conformational change in the RAE tandem of ELMO2 upon binding to BAI1-EBD. Except the N-terminal 4 residues and the C-terminal 7 residues, the electron densities of the rest of the BAI1-EBD (i.e., aa 1471-1495) are clearly defined in the complex (Supplementary Fig. 4). The BAI1-EBD peptide, composed of two short α-helices, occupies the elongated concave

groove formed by the ARR and ELMO domains (Fig. 4a, b). Interestingly, the ARMs together with a folded domain serve as a functional unit for target recognition. Notably, the binding site of BAI1 on ELMO2 is far away from the RhoG-binding site (Supplementary Fig. 5a), suggesting that RhoG and BAI1 would not compete with each other when binding to ELMO2-RAE. Indeed, addition of excess amount of the BAI1-EBD peptide did not affect the binding between RhoG and ELMO2-RAE (Supplementary Fig. 5b).

**Interfaces between ELMO2-RAE and BAI1-EBD**. The high-resolution structure of the complex provides the opportunity to dissect the binding details between the two proteins. The exquisite specific interfaces on ELMO2-RAE are formed by the inner α-helices of ARM2-5 of ARR domain and the 16-residue flexible loop connecting α6 and α7 of ELMO domain (Fig. 4a). The complex assembly is mainly mediated by two parts of interactions: (1) the C-terminus of BAI1-EBD forms extensive polar interactions with ELMO2-RAE. For example, R1489$^{BAI1}$ forms a salt bridge with E198$^{ELMO2}$; H1485$^{BAI1}$ forms hydrogen bonds with E281$^{ELMO2}$ and E429$^{ELMO2}$, and the sidechain of H284$^{ELMO2}$ forms hydrogen bonds with the mainchain of I1483$^{BAI1}$ and sidechain of N432$^{ELMO2}$ (Fig. 4c); (2) the N-terminus of BAI1-EBD mainly forms hydrophobic interactions with ELMO2-RAE. The hydrophobic core is formed by Tyr1475, Leu1478, Phe1480, Ile1483, Met1484 from BAI1-EBD and Leu202, Leu244, Val288, Leu292, Pro427 from ELMO2-RAE (Fig. 4d). There are also polar contacts in this interface. Particularly, S1473$^{BAI1}$, R1474$^{BAI1}$, E248$^{ELMO2}$, R251$^{ELMO2}$, E298$^{ELMO2}$ form a hydrogen bond network (Fig. 4d). Importantly, the residues that contribute to the binding interfaces are highly conserved among BAI and ELMO family proteins from different species (Fig. 1a and Supplementary Fig. 3), implicating the indispensable functions of the BAI/ELMO interactions during evolution.

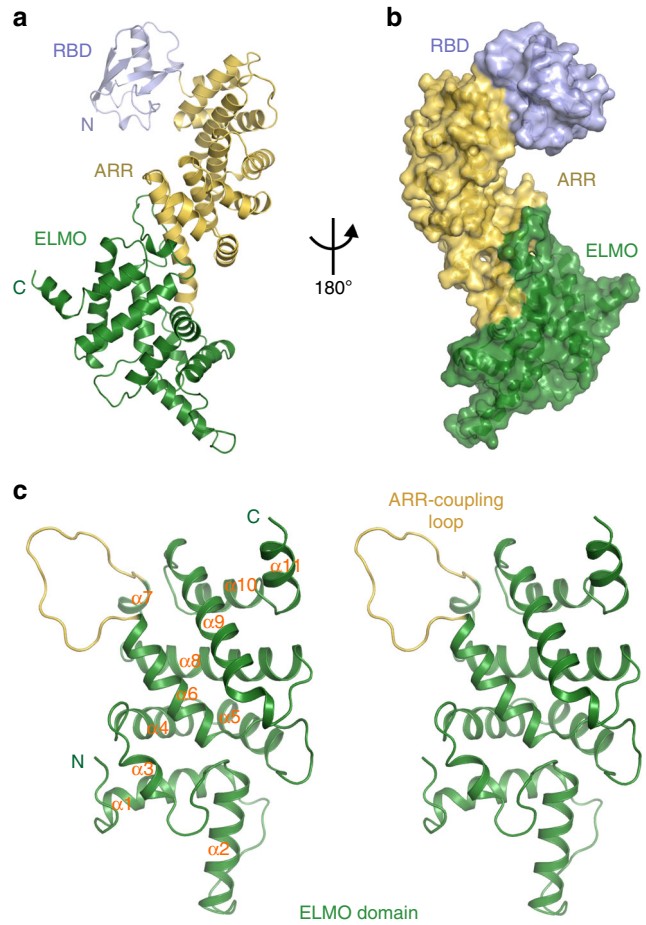

**Fig. 2** Overall structure of the ELMO2-RAE tandem. **a** Ribbon diagram of overall structure of the RAE tandem of ELMO2. The RBD, ARR, ELMO domains are colored in blue, yellow, and green, respectively. **b** Surface representation showing the overall architecture of the RAE tandem with the same color scheme as in panel **a**. **c** Stereo-view of the structure of the ELMO domain. Noted that a flexible loop located between α6 and α7 of the ELMO domain is required for the ARR domain binding

To verify whether the key residues involved in the interfaces are important for complex assembly in solution, we made a series of mutations on ELMO2-RAE and BAI1-EBD and tested their binding capacities using GST-pull down assays. The mutants of ELMO2-RAE behaved well in solution and did not induce significant conformational changes of the overall structure (Supplementary Fig. 6). As expected, mutations of key residues that contribute to the interfaces all significantly impaired the BAI1-EBD/ELMO2-RAE interaction (Fig. 4e, f). Specifically, substitution of R1489$^{BAI1}$ with Ala totally diminished its binding to ELMO2-RAE most likely due to the disruption of the electrostatic interaction of R1489$^{BAI1}$-E198$^{ELMO2}$ pair (Fig. 4e), explaining why replacement of three positive charged residues (i.e., $^{1487}$RKR$^{1489}$ in mBAI1, Fig. 1a) with triple Ala or Glu in BAIs significantly reduced the BAI/ELMO bindings and affected the functions of the complex in phagocytosis and myoblast fusion[14,17].

**BAI-binding deficient ELMO2 mutant impairs myoblast fusion.** Next, we wanted to use myoblast fusion assay to examine the functional relevance of the BAI/ELMO binding interfaces in vivo. In the cultured C2C12 cells, loss of BAI3 severely impaired myoblast fusion and can be rescued by the wild-type BAI3, but not the BAI3 mutant deficient in ELMO binding. Loss

of ELMO2 displayed similar defects in myoblast fusion[17]. Consistent with this, quantification of the percentage of myonuclei in myotubes with different sizes showed that knockdown of ELMO2 drastically affected the formation of the large myotubes with more than four nuclei in MHC positive C2C12 cells (Fig. 4g, h and Supplementary Fig. 7a), indicating a specific blockade to myoblast fusion independent of muscle differentiation. Expression of the wild-type ELMO2 significantly rescued the fusion defects caused by the loss of endogenous ELMO2. However, the BAI-binding deficient ELMO2$^{E198A}$ failed to restore myoblast fusion (Fig. 4f–h and Supplementary Fig. 7b). These data clearly demonstrated that the BAI/ELMO binding interfaces revealed by the complex structure are indispensable for their cellular functions.

**Disease-causing mutations in the BAI1/ELMO2 interface.** Given the essential roles of BAIs in diverse cellular processes, it would not be surprising that BAIs have been implicated in numerous diseases such as cancers, immunological disorders, psychiatric disorders and neurological diseases[6,8,10,18]. The atomic structure of BAI1/ELMO2 complex allow us to dissect the molecular basis of the diseases caused by these two genes. A number of somatic mutations of BAI1 have been found in patients with various cancers[28]. For example, the p.M1486K$^{BAI1}$, p.Y1477C$^{BAI1}$, p.L1422H$^{BAI3}$ and p.V1427G$^{BAI3}$ mutants were all found in the patients with lung cancers[29–31] (Fig. 5a). The four corresponding residues in mouse BAI1 (i.e., M1484$^{mBAI1}$, Y1475$^{mBAI1}$, L1478$^{mBAI1}$, and I1483$^{mBAI1}$), which are located at the second binding interface, form hydrophobic contacts with Leu202, Leu244, Val288, Leu292, Pro427 from ELMO2-RAE (Fig. 5b). Satisfyingly, the p.M1486K$^{BAI1}$, p.L1422H$^{BAI3}$ and p.V1427G$^{BAI3}$ mutants showed undetectable bindings to ELMO2-RAE (Fig. 5c, d). To our surprise, the p.Y1477C$^{BAI1}$ mutant still retained the binding capacity to ELMO2-RAE (Fig. 5c), probably due to the relative strong hydrophobicity of the sidechain of cysteine. Future work is required to dissect whether these cancer-associated mutations induce tumorigenesis via interfering with the BAI/ELMO interactions in the physiological conditions.

ELMO2 was associated with intraosseous vascular malformation (VMOS), a disease characterized with non-neoplastic expansions of blood vessels due to errors during angiogenesis[32]. A deletion mutant of ELMO2, p.Ala311_Thr355del, was found in VMOS patients and caused a significant decreased level of DOCK1, thus resulting in deficient Rac1-depedent cell migration[32]. Based on our structures, the deletion mutant is expected to impair the RAE supramodular assembly of ELMO2, thus affecting the binding(s) of ELMO2-RAE with its target(s)(such as BAIs).

**GPR128 as another aGPCR upstream of ELMO/DOCK/Rac axis.** We next wanted to know whether there exists other aGPCR(s) that orchestrate(s) the intracellular signaling pathways via the ELMO/DOCK/Rac module. We performed a detailed amino acid sequence analysis of the cytosolic tail of all aGPCRs, and found that GPR128 (also known as ADGRG7), an aGPCR highly expressed in intestinal tissues[33], contains a C-terminal fragment (aa 750-778) displaying high sequence similarity to BAI1-EBD (Fig. 6a, b). More importantly, the majority of residues corresponding to those involved in the ELMO2 binding of BAI1-EBD are conserved in GPR128 (Fig. 6b). Satisfyingly, ELMO2-RAE, but not RBD-ARR tandem, bound to the intact cytoplasmic tail of GPR128 (GPR128-CT, aa 717-785) effectively in a GST-pull down-based assay (Fig. 6c), suggesting that the binding mode of ELMO2/GPR128 complex is most likely the same as that of ELMO2/BAI1 interaction. Moreover, the RAE tandem of ELMO1 and ELMO3 both bound to GPR128-CT as effectively as

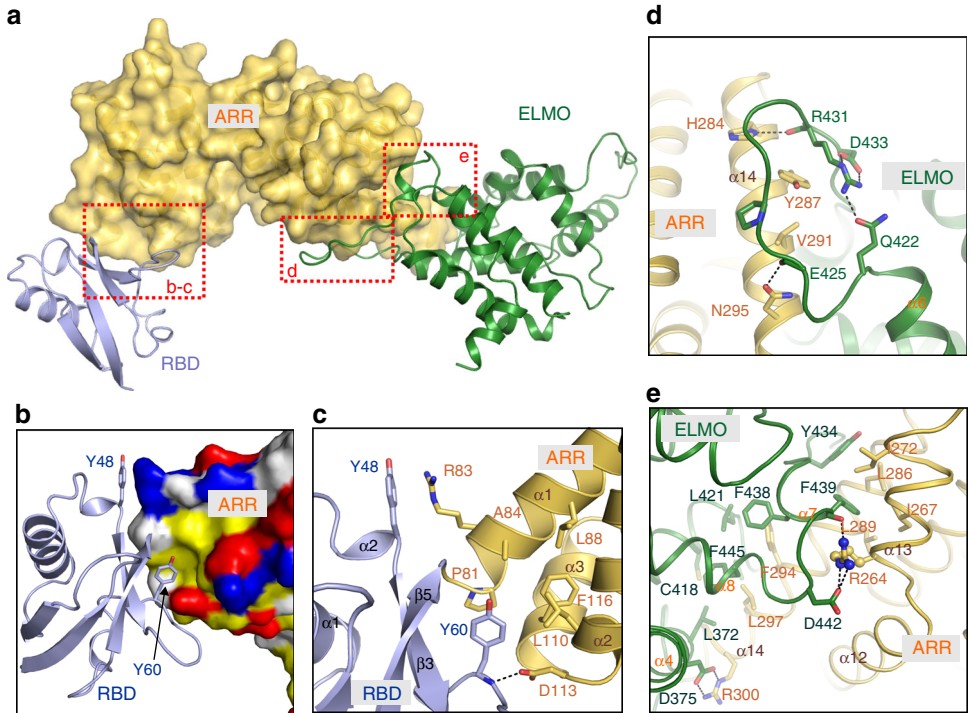

**Fig. 3** The binding interfaces between the ELMO2-RAE tandem. **a** The combined surface and ribbon representations of the ELMO2-RAE tandem. Noted that the RBD, ARR, and ELMO domains contact together forming a structural supramodule. **b** The combined electronical surface and ribbon representations of the RBD/ARR interface showing that Y60$^{RBD}$ inserts into a hydrophobic pocket of the ARR domain. **c** Detailed interface of the RBD/ARR binding site. **d** Detailed interface of the binding site between the ARR domain and the flexible loop located between α7 and α8 of the ELMO domain. **e** Detailed interface of the second ARR/ELMO binding site. Noted that this interface is mainly mediated by hydrophobic interactions. Dotted lines represent hydrogen bonds

ELMO2-RAE did (Supplementary Fig. 8). To further verify the binding mode, we chose to mutate the Arg772 of GPR128 (the corresponding residue in BAI1 is Arg1489 whose mutation (R1489A) would disrupt the BAI1/ELMO2 interaction, Fig. 6b and Fig. 4e) into Ala, and tested its binding capacity to ELMO2-RAE. As expected, the R772A$^{GPR128}$ mutant showed diminished binding to ELMO2-RAE (Fig. 6d). Additionally, substitution of Tyr760 (the corresponding residue in BAI1 is Leu1478, which involves in the hydrophobic interface of BAI1/ELMO2, Fig. 4d) with Asp also eliminated the GPR128/ELMO2 interaction (Fig. 6d). These data further indicated that the GPR128/ELMO2 complex also employs the binding mode uncovered by the BAI1/ELMO2 complex structure.

## Discussion

Despite that the aGPCR family is widely linked to enormous important physiological processes and a variety of human diseases, little is known about molecular basis of aGPCR-mediated intracellular signaling. No drugs have been approved to target any members of these GPCR family. Emerging evidence showed that, in addition to canonical G protein signaling, majority of aGPCRs could trigger cellular signaling via the G protein-independent pathways. Thus, elucidation of the molecular mechanism governing the aGPCR-mediated intracellular signaling is important for potential drug discovery. A comprehensive sequence analysis of the C-terminal cytoplasmic tails of all aGPCRs revealed that many of aGPCR members (including BAIs) possess a PDZ domain-binding motif (PBM) at their very C-terminus (Supplemental Fig. 9), suggesting that the PDZ/PBM interaction may be critical for the intracellular functions of these aGPCRs. Consistently, BAI1 was found to interact with the PDZ domain of Rac-GEF Tiam1 which functions together with the polarity factor

Par3 to regulate Rac signaling during synaptogenesis in hippocampal neurons[21].

In addition to the PBM-mediated signaling pathways, the BAI-aGPCRs exert their cellular functions through the ELMO/DOCK/Rac axis. In this study, we performed systematically biochemical and structural studies of the BAI1/ELMO2 interaction. Our study reveals that the RBD-ARR-ELMO (RAE) supramodule of the ELMO family scaffolds specifically recognizes a conserved C-terminal fragment of the BAI family aGPCRs. It is well known that the ELMO/DOCK/Rac complex acts downstream of RhoG to regulate diverse signaling pathways. Intriguingly, we found that binding of BAI1 to ELMO2 would not affect RhoG/ELMO2 interaction, suggesting that the RhoG/ELMO/DOCK pathway and the BAI/ELMO/DOCK pathway may cooperate with each other in orchestrating the Rac-dependent actin cytoskeleton dynamics.

An interesting finding in the complex structure is that, the RAE tandem, but not the ARR domain alone, forms a right-handed superhelical architecture and creates a concave groove responsible for BAI1 binding. This unique ARR/target recognition mode is unlike the typical mode observed in most ARM-containing proteins such as β-catenin, APC and importin-α (Supplementary Fig. 10). These observations imply that the folded domains N- or C-terminal to the canonical ARMs would form a structural and functional supramodule together with the ARR domain, which provides further binding specificity toward the target(s).

Another important discovery of this work is that, based on the BAI1/ELMO2 structure, we identified GPR128 as another upstream aGPCR for the ELMO/DOCK/Rac signaling module. Notably, the action mode of GPR128 via ELMO scaffolds is most likely the same as that of BAI1/ELMO2 assembly demonstrated in our complex structure. GPR128 is phylogenetically related to GPR56[34]. Loss of GPR128 led to reduced body weight gain and

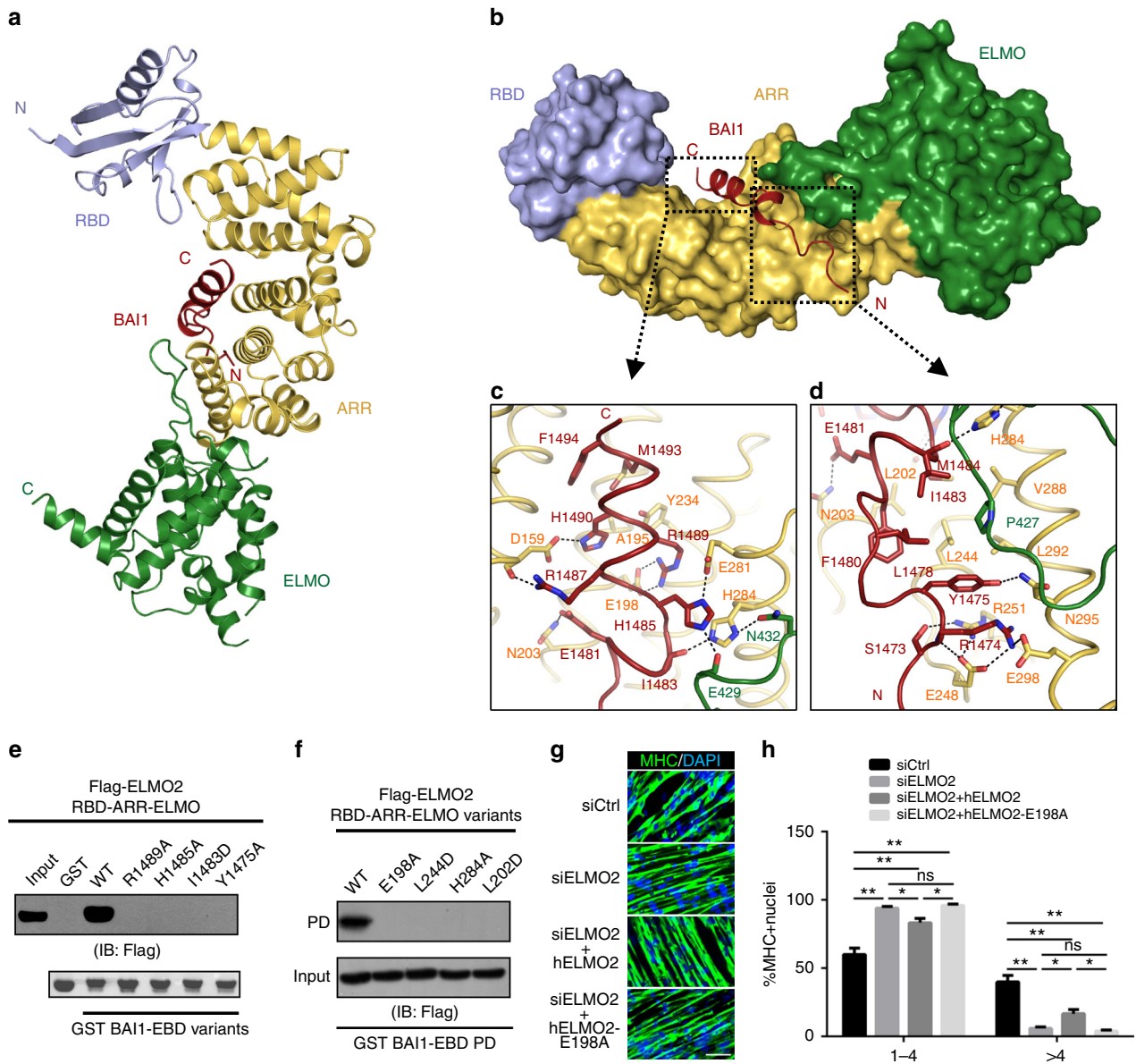

**Fig. 4** The crystal structure of the BAI1/ELMO2 complex. **a** Ribbon diagram of overall structure of the ELMO2-RAE/BAI1-EBD complex. **b** The combined surface and ribbon representations of the ELMO2-RAE/BAI1-EBD complex structure. **c**, **d** Detailed interactions between ELMO2-RAE and BAI1-EBD. **e**, **f** GST-pull down-based assays showing that key residues involved in the ELMO2-RAE/BAI1-EBD interface are required for the intact interaction. PD, pull down. **g** Representative images of myoblast fusion showing that down-regulation of ELMO2 impairs the formation of the large myotubes with more than four nuclei in MHC positive C2C12 cells. The wild-type ELMO2, but not the BAI-binding deficient ELMO2-E198A mutant, significantly rescued the myoblast fusion. Scale bar, 50 μm. **h** Quantification of experiments shown in **g**. Three independent batches of cultures were imaged for each group ($n = 1000$) for quantifications. All the results are presented as mean ± SEM. *$p < 0.05$, **$p < 0.01$, ***$p < 0.001$, ****$p < 0.0001$

increased intestinal contraction frequency[33]. Rac signaling is well-known for its functions in actin cytoskeleton remodeling, cell motility and migration[35]. It is tempting to speculate that the GPR128/ELMO/DOCK/Rac axis might play a role in the intestinal motility. Future investigations are needed to explore whether the GPR128/ELMO-mediated signaling is indeed involved in the physiological environment.

## Methods
**Protein expression and purification.** The coding sequences of the RBD-ARR-ELMO (RAE) tandem of ELMO2 (aa 1-520) and various fragments of BAI1 cytosolic tail were amplified from the *ELMO2* gene (GenBank: AF398886.1) and *BAI1* gene (GenBank:NM_174991.3) (Supplementary Table 1), respectively. All of

the mutations were introduced by the standard PCR-based mutagenesis method using the Phanta Max superfidelity DNA polymerase (Vazyme Biotech Co., Ltd., catalogue no. P505) and confirmed by DNA sequencing. Fragments were cloned into a modified pET-32a vector (with Trx-His$_6$-tag) or pGEX4T-1 vector (with GST-tag) and expressed in BL21 (DE3) *E. coli* cells for 18 h at 16 °C inducing by the isopropyl-β-D-thiogalactoside (IPTG) at a final concentration of 0.2 mM. The recombinant proteins were purified using Ni$^{2+}$-NTA agarose affinity chromatography or GSH-Sepharose affinity chromatography (GE Healthcare) followed by the size-exclusion chromatography (SEC) in the buffer containing 50 mM Tris pH 8.0, 100 mM NaCl, 1 mM EDTA, and 1 mM DTT.

For the crystallography, the Trx-His$_6$-ELMO2-RAE was further cleaved by the human rhinovirus 3 C protease and the Trx-His$_6$ tag was removed by another step of SEC in the buffer containing 25 mM HEPES pH 7.4, 150 mM KCl, 10% glycerol. Selenomethionine-labeled ELMO2-RAE was expressed in *E. coli* B834 (DE3) cells using M9 minimal media and purified using the same procedure as described above for the native protein.

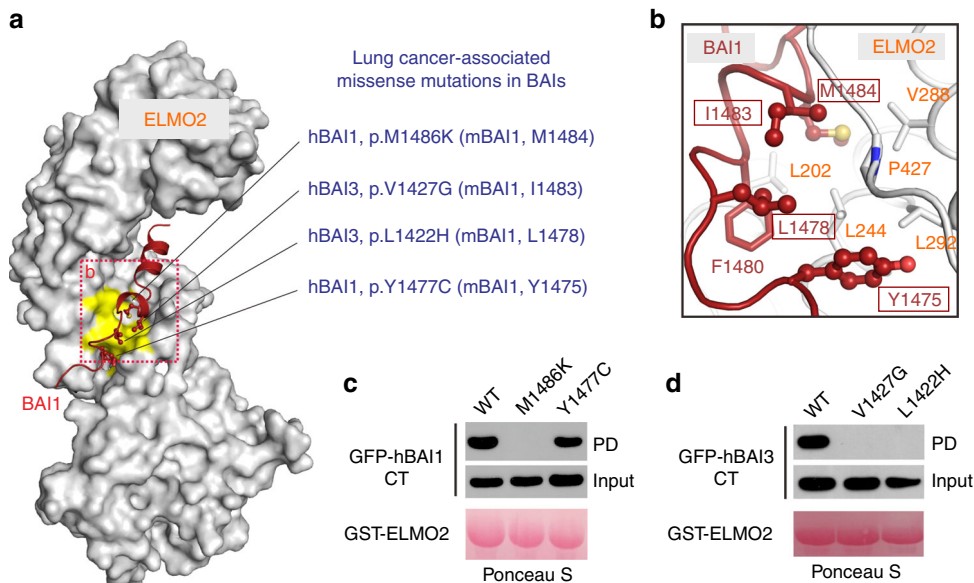

**Fig. 5** Cancer-associated mutations in BAI/ELMO interface. **a** Four lung cancer-associated missense mutations in BAIs mapped on to the interface of the complex structure. In this drawing, the side chains of key residues associated with cancers are shown in the stick mode, and the residues on ELMO2 that coordinate these cancer-related residues are color in yellow. **b** Zoomed-in view of binding details of cancer-related residues shown in panel **a**. **c**, **d** Effects of the cancer-associated mutations on the formations of the BAI1-CT/ELMO2-RAE (**c**) and BAI3-CT/ELMO2-RAE (**d**) complexes based on the GST-pull down assays

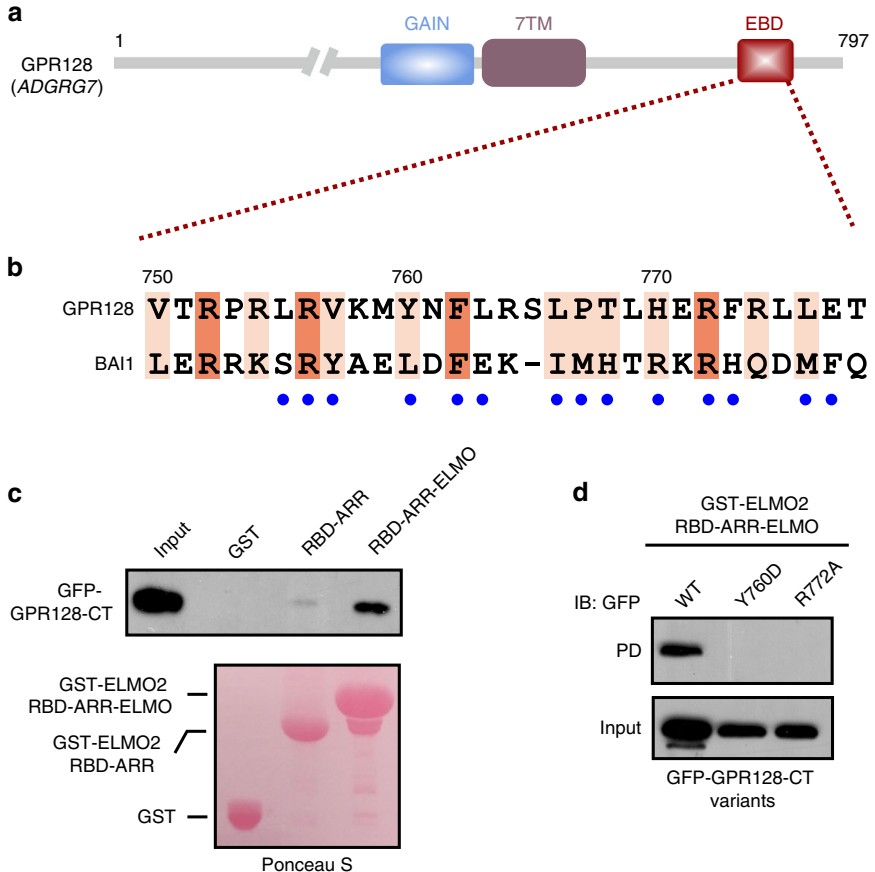

**Fig. 6** GPR128 as another aGPCR upstream of ELMO family scaffolds. **a** Domain organization of GPR128. **b** Sequence alignment of the ELMO-binding domain from GPR128 and BAI1. In this alignment, the absolutely conserved and conserved residues are highlighted with orange and pink boxes, respectively. Residues of BAI1 involved in binding to ELMO2 are annotated below as blue dots. **c** GST-pull down assays showing that ELMO2-RAE, but not RBD-ARR tandem, specifically binds to GPR128-CT. **d** Mutation of residues of GPR128-CT corresponding to those of BAI1 involved in ELMO2 binding blocked the GPR128/ELMO2 interaction in the GST-pull down assays. PD, pull down

**Isothermal titration calorimetry assay**. ITC measurement was carried out on a MicroCal iTC200 system (Malvern Panalytical, UK) at 25 °C. Both of ELMO2 and BAI1 proteins were in the buffer containing 50 mM Tris pH 8.0, 100 mM NaCl, 1 mM EDTA, and 1 mM DTT. The BAI1-EBD (~500 μM) was loaded into the syringe and injected 2 μl aliquot into the cell placed with the ELMO2-RAE (~40 μM) in each titration. The time interval was 120 s in each titration to make sure that the titration peak returned to the baseline. The experiment data were analyzed by the program of Origin7.0 (Microcal).

**GST pull-down assays**. Various wild-type or mutants of Flag-tagged ELMO2 proteins or BAI proteins were overexpressed in HEK293T cells. Cells were harvested after 24 h transfection and lysed with ice-cold lysis buffer (50 mM HEPES [pH 7.4], 150 mM NaCl, 1 mM EGTA, 10% glycerol, 1% Triton and protease inhibitor cocktail) at 4 °C for 1 h. After centrifugation, the supernatants were incubated with GST-tagged BAIs or ELMO2 proteins pre-loaded to 30 μl GSH-sepharose 4B slurry beads in the lysis buffer at 4 °C for 1 h. After extensive washing, the captured proteins were eluted by boiling with 20 μl 2× SDS-PAGE loading dye and detected by western blotting with the anti-Flag antibody (Sigma; catalogue no. F1804; 1:3000). GFP-tagged GPR128 cytosolic tail (CT) (GenBank: NP_766413.2, aa 717-785) and various GPR128-CT mutants were also overexpressed in HEK293T cells. The pulldown assays between various fragments of GPR128-CT and GST-tagged ELMO2-RAE were performed using the method described above. The bound proteins were detected by western blotting with the anti-GFP antibody (Santa Cruze; catalogue no. sc-9996; 1:10000).

**Crystallization and Structure determination**. Crystals of both native and Se-Met ELMO2-RAE were grown in 0.1 M Sodium Malonate pH 4.0, 12% PEG3,350 by the sitting-drop vapor diffusion method with drops consisted of 0.5 μL protein (10 mg/ml) and 0.5 μL reservoir solution at 16°C. Crystals were cryoprotected by adding ethylene glycol into the mother liquor to the final concentration of 20% (v/v) and quickly frozen into liquid nitrogen. Diffraction data for both native and Set-Met crystals were collected at the Shanghai Synchrotron Radiation Facility (SSRF, BL18U, China) at 100 K with a wavelength of 0.97853 Å and processed by HKL3000 package[36]. Phases were calculated form a SAD dataset using the PHE-NIX program[37]. The structures were build using Coot[38] and refined with PHENIX[37].

For the reconstitution of the ELMO2-RAE/BAI1-EBD complex, ELMO2-RAE was mixed with a commercial synthetic BAI1-EBD peptide (ChinaPeptide Co.,Ltd.) in a molar ratio of 1:2. The best crystals of the ELMO2-RAE/BAI1-EBD complex (10 mg/ml) were grown in 0.3 M Ammonium Fluoride, 18% PEG3,350 by the sitting-drop vapor diffusion method at 16°C. Using the model of ELMO2-RAE as the template, the structure of ELMO2-RAE/BAI1-EBD complex was solved by the molecular replacement method with Phaser[39]. The BAI1-EBD peptide was manually built. Further model buildings and refinement were carried out using Coot[38] and PHENIX[37] alternately. The final refinement statistics of the structures of ELMO2-RAE and its complex with BAI1-EBD are summarized in Table 1. Structural diagrams were prepared by Pymol[40].

**Cell culture and Immunofluorescent staining**. C2C12 cells were cultured in growth medium (DMEM supplied with 20% (v/v) FBS) and induced to differentiate by switching to differentiation medium (DMEM supplied with 2% (v/v) horse serum). For knockdown of ELMO2, C2C12 cells were transfected with the short interfering RNA (siRNA) targeting mouse ELMO2 (siELMO2#1: 5'-GGCCT TCTCCATCCTGTAT-3'; siELMO2#2: 5'- GGCTCAGAGAGACATTATA-3') using Lipofectamine RNAiMAX (Invitrogen). Gene knockdown efficiency was assessed by quantitative real-time PCR (Q-RT-PCR). Total RNA was purified from C2C12 cells using the Trizol reagent (Invitrogen). Reverse transcription was carried out with the ImProm-II Reverse Transcription System (Promega). RT-PCR was performed using corresponding specific primers (ELMO2_Forward: 5'-TCACCA AGATGGATCCCAAT-3'; ELMO2_Reverse: 5'- GGAGTCTGGGTGAAGTCCA A-3') in a LightCycler 480 Instrument II (Roche) with the SYBR Green System (Roche). The data was analyzed using the comparative Ct method with the expression of GAPDH as a control.

For the rescue experiments, $5 \times 10^5$ 293 A cells were transfected with 1 μg pAd/ BLOCK-iT™-DEST (Thermo Fisher Scientific) vector carrying Flag-tagged full length human ELMO2 and ELMO2-E198A mutant. Cells containing adenovirus were harvested after 10 days and lysed. Adenovirus was added into the C2C12 cells together with siRNAs, followed by further immunofluorescent staining or western blotting (Sigma; catalogue no. F1804; 1:3000).

For the immunofluorescent staining, C2C12 cells were fixed in 4% polyformaldehyde (PFA) in PBS for 5 min, and permeabilized with 1% Triton in PBS (PBST), and then blocked in PBST containing 4% IgG-free BSA for 1 h. After extensive washing with PBS, cells were incubated with the primary antibody targeting MHC (Developmental Studies Hybridoma Bank; catalogue no. MF20; 1:200) at 4 °C overnight. Then, cells were washed with PBS and incubated with an Alexa Fluor 488 conjugated donkey anti-mouse secondary antibody (Invitrogen; catalogue no. A-21202; 1:1000;) for 1 h. DAPI (Sigma; catalogue no. D9542; 1:1000) was used to reveal nuclei. Images were taken with a RT3 CCD Camera (SPOT).

## Data availability

The atomic coordinates of ELMO2-RAE and the ELMO-RAE/BAI1-EBD complex have been deposited to the Protein Data Bank under the accession codes 6IE1 and 6IDX, respectively. Other data are available from the corresponding author upon reasonable request. A reporting summary for this article is available as a Supplementary Information file. The source data underlying Figs. 1c–e, 4e–f, 5c, d, 6c, d and Supplementary Figs. 1c, 5b and 8 are provided as a Source Data file.

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

## Acknowledgements

We thank the Shanghai Synchrotron Radiation Facility (SSRF, China) BL18U for X-ray beam time, the staff members of the Large-scale Protein Preparation System and Molecular Imaging System at the National Facility for Protein Science in Shanghai (NFPS), Zhangjiang Lab, China for providing technical support and assistance in data collection and analysis, Dr. Deqiang Yao (Shanghai Jiao Tong University, School of Medicine) and Dr. Yuan Shang (University of Arizona) for their technical support during structure determination, Prof. Mingjie Zhang (Hong Kong University of Science and Technology) for valuable discussion and critical comments on the manuscript. This work was supported by grants from the National Key R&D Program of China (2017YFA0504901) to R.Z., the Strategic Priority Research Program of the Chinese Academy of Sciences (XDB08030104) to R.Z., the National Key R&D Program of China (2018YFA0507900) to J.Z., the Chief Scientist Program of Shanghai Institutes for Biological Sciences, Chinese Academy of Sciences to R.Z., and the National Natural Science Foundation of China (31770779, U1532121, 31470733) to J.Z.

## Author contributions

Z.F.W., J.Z. and R.Z. designed the experiments. Z.F.W. performed the biochemical and structural experiments. L.L. provided technical support and assistance in structure determination and analysis, C.S. performed myoblast fusion assays. Z.G.W., R.Z. and J.Z. analyzed the data. J.Z. wrote the paper. J.Z. coordinated the project. All authors approved the final version of the manuscript.

## Additional information

**Competing interests:** The authors declare no competing interests.

