## [Peer Review File · Nature Communications]

Reviewers' comments:

Reviewer #1 (Remarks to the Author):

In this work, Weng et al determine the 3D structure of the ELMO2 domains, specifically the RBD-Armadillo repeats-ELMO domain region (RAE), followed by the complex of BAI1 helix peptide bound to the RAE-ELMO2 protein. Although the interaction between ELMO1,2,3 proteins and the BAI1,2,3, proteins was detailed by Park et al many years ago, the structure of this interaction was never solved and given the recent renewed interest in BAI1 and particularly adhesion GPCRs, this becomes particularly important to define. In this work the authors initially confirm the observations of Park et al that a helical region within the tail of BAI1 binds to ELMO2, and then went on to solve the RAE structure. Here they nicely define a novel interaction between the ARR and the ELMO domains and this led to the identification of a potential groove formed via the two regions. Further, when they solved the 3D structure of the complex, the authors nicely show that the BAI1 helical peptide (actually two short alpha helices) make multiple contacts with the ARR and ELMO domains. Further, they identify the specific residues on both BAI1 (including the arginine part of the three residues originally mutated by Park et al) and the residues on the two domains of ELMO2 that could mediate the binding. They also perform additional mutagenesis, in the myoblast fusion based assay and based on the mutations within BAI1 seen in lung cancers, to confirm the structural studies. Lastly, the authors go on to identify GPR128 as another adhesion GPCR that could potentially be a binding partner of ELMO2.

In sum, this is a beautiful piece of work that solves the structure of BAI1-Elmo2 complex and the authors should be commended for the thorough and insightful series of observations. Because this is already a thorough paper, I have only minor comments that could simply be answered via text edits or if the authors have experiments, this would be useful to provide (but not required).

1. Have the authors modeled the structures of ELMO1 against ELMO2 and 3 – how do they compare? Since they all bind efficiently to BAI1 and BAI3?
2. Have the authors ever modeled the RhoG on the ELMO-RBD in the context of this structure? They comment that the ARR-ELMO domain regions will not likely affect the RBD, and that RhoG could likely still bind, but if they can add further insights on this point, this would be useful for the readers.

Reviewer #2 (Remarks to the Author):

Adhesion GPCRs are a novel GPCR family with very large extracellular and intracellular regions. Emerging studies link them to various cellular processes. While there is increasing attention on the structures and functions of the extracellular regions, the structures and interactions of the intracellular regions has lagged behind. Although G-protein coupling occurs via the cytoplasmic tails, the very large intracellular regions should be involved in other functions.

This manuscript provides a first structure/function analysis of the intracellular interactions of the important subfamily of BAI proteins. Weng et al. found an evolutionarily conserved fragment located in the intracellular region of BAI1 specifically binds to the RBD-ARR-ELMO (RAE) tandem of ELMO2 via biochemical characterization of BAI/ELMO2 interaction. ITC results show that BAI1-EBD binds to ELMO2 in a 1:1 molar ratio. The authors then solved the crystal structures of the RAE tandem of ELMO2 and its complex with conserved BAI1 intracellular region, respectively. The structures indicated that the RBD, ARR, ELMO domains of ELMO2 form a structural and functional supramodule required for BAI1 binding. Further structural analysis revealed the interfaces between ELMO2-RAE and BAI1-EBD. On a functional level, the authors used myoblast fusion assay to examine the functional relevance of the BAI/ELMO binding interfaces. The results show that mutants deficient in binding failed to restore myoblast fusion. Finally, amino acid sequence analysis found that a GPR128 C-terminal fragment displaying high sequence similarity to BAI1-

EBD, and the GPR128/ELMO2 complex also employs the binding mode uncovered by the BAI1/ELMO2 complex structure.

Whether this interaction can be generalized for all adhesion GPCRs or not, this study will be a first to attract the needed attention to the unknown functions of the intracellular regions of adhesion GPCRTs. The structural and functional work is decent and I recommend publication after the below points are addressed.

Abstract:

1) "The adhesion G protein-coupled receptors (aGPCRs) play crucial roles in diverse cellular processes through the G protein-independent signaling pathways, though the underlying molecular mechanism is not well understood."

Please revise. Not all G-protein independent pathways are TM dependent. There are TM independent functions of adhesion GPCRs which are unrelated to the current study. Please be specific and cite such functions in the intro.

2) "...another aGPCR-GPR128 was identified as a new upstream receptor for the ELMO family scaffolds, most likely with a recognition mode similar to that of BAI/ELMO pairs, thus suggesting a general action mode for aGPCRs. "

The word general mechanism should be used carefully. 2 out of 33 adhesion GPCRs does not qualify for general.

Other:

3) Please use better grammar.

4) A comprehensive sequence analysis of the intracellular regions of all adhesion GPCRs will be a great supplement for this paper.

5) Is the ELMO binding peptide at the very C-terminus? How many other residues until the end of the C-terminus. Some adhesion GPCRs have been proposed to have PDZ binding sites at the C-terminus. A general analysis will be great.

6) Page 6: Typo (aa 1167-1502)

7) Use of the word coupling interfaces should be clarified. Binding will be a better word.

8) The proper expression and folding of the BAI mutants and ELMO mutants is critical for the validity of the experiments that report a mutation abolishes binding. A mutation that unfolds the protein will also give the same result although it is not at the binding interface. Please provide expression/folding information.

9) How many ELMOs are there and which ELMO is used? GPR128 looks like it binds to ELMO less. Can it bind to other ELMOs more?

10) GPR128-CT in Fig 6. Is this full-length CT?

11) Does BAI-EBD bind to G protein? If it does, how BAI intracellular region choose different binding partners and influence the signal transduction?

12) Why some BAI-EBD variants show as multiple bands on the gel, such as "H1485A" in figure. 4, panel e.

Point-by-point responses to the comments and suggestions raised by the reviewers

(Comments from the reviewers in *black*, and our responses in *blue*)

Reviewer #1 (Remarks to the Author):

In this work, Weng et al determine the 3D structure of the ELMO2 domains, specifically the RBD-Armadillo repeats-ELMO domain region (RAE), followed by the complex of BAI1 helix peptide bound to the RAE-ELMO2 protein. Although the interaction between ELMO1,2,3 proteins and the BAI1,2,3, proteins was detailed by Park et al many years ago, the structure of this interaction was never solved and given the recent renewed interest in BAI1 and particularly adhesion GPCRs, this becomes particularly important to define. In this work the authors initially confirm the observations of Park et al that a helical region within the tail of BAI1 binds to ELMO2, and then went on to solve the RAE structure. Here they nicely define a novel interaction between the ARR and the ELMO domains and this led to the identification of a potential groove formed via the two regions. Further, when they solved the 3D structure of the complex, the authors nicely show that the BAI1 helical peptide (actually two short alpha helices) make multiple contacts with the ARR and ELMO domains. Further, they identify the specific residues on both BAI1 (including the arginine part of the three residues originally mutated by Park et al) and the residues on the two domains of ELMO2 that could mediate the binding. They also perform additional mutagenesis, in the myoblast fusion based assay and based on the mutations within BAI1 seen in lung cancers, to confirm the structural studies. Lastly, the authors go on to identify GPR128 as another adhesion GPCR that could potentially be a binding partner of ELMO2.

In sum, this is a beautiful piece of work that solves the structure of BAI1-Elmo2 complex and the authors should be commended for the thorough and insightful series of observations. Because this is already a thorough paper, I have only minor comments that could simply be answered via text edits or if the authors have experiments, this would be useful to provide (but not required).

We thank the reviewer for the positive comments on our work.

1. Have the authors modeled the structures of ELMO1 against ELMO2 and 3 – how do they compare? Since they all bind efficiently to BAI1 and BAI3?

Following the reviewer's suggestion, the structures of RBD-ARR-ELMO (RAE) tandems of ELMO1 and ELMO3 were modeled using the structure of ELMO2-RAE as a template. As expected, the overall structures of ELMO1-RAE and ELMO3-RAE were very similar to that of ELMO2-RAE (**FIG-I-A**). Further surface electrostatic potential analysis, together with the sequence alignment shown in the **Supplemental Figure 3**, demonstrated that the BAI-binding sites on ELMO1, 2, 3 are similar (**FIG-I-B-D**), suggesting that all the members of ELMO family scaffolds could bind to BAI-aGPCRs via the same interaction interface.

FIG-I. Structural comparison of RAE tandems from ELMO1, ELMO2, and ELMO3. (A) Superposition of the structures of ELMO1-RAE (modeled structure), ELMO2-RAE (this work), and ELMO3-RAE (modeled structure). (**B-D**) Electrostatic potential mapped onto the

molecular surface of ELMO1-RAE (**B**), ELMO2-RAE (**C**), and ELMO3-RAE (**D**). The BAI1-binding sites on ELMOs are indicated with a black dash circle.

2. Have the authors ever modeled the RhoG on the ELMO-RBD in the context of this structure? They comment that the ARR-ELMO domain regions will not likely affect the RBD, and that RhoG could likely still bind, but if they can add further insights on this point, this would be useful for the readers.

This is an excellent point! Following the reviewer's suggestion, we have made the homology modelling of RhoG/ELMO2/BAI1 complex. Homology models of RhoG and RhoG/ELMO2 complex were generated using Rac1-GTP (PDB code: 1E96) and Ras/RalGDS complex (PDB code: 1LFD) as the template, respectively (**FIG-II-A**). In the model structure, RhoG and BAI1 bind to different sites on ELMO2, suggesting that binding of BAI1 to ELMO2 would not affect the RhoG/ELMO2 interaction. Indeed, in a competition pull down assay, addition of excess amount of BAI1 peptide did not affect the binding between RhoG and ELMO2-RAE (**FIG-II-B**). We have included these points in the revised manuscript (**revised Fig. S5**).

FIG-II (revised Fig. S5). Binding of BAI1 to ELMO2 didn't affect RhoG/ELMO2 interaction. (A) Homology model of RhoG/ELMO2/BAI1 complex based on the Ras/RalGDS complex structure (PDB code: 1LFD). RhoG, ELMO2-RAE and BAI1 peptide are color in green, gray, and red, respectively. Noted that the binding site of BAI1 on ELMO2 is far away from the RhoG-binding site. (B) Pull down assays showing that RhoG bound to ELMO2-RAE effectively with (*lane 3*) or without (*lane 2*) BAI1 peptide. In the assays, the concentrations of GST-ELMO2-RAE and the BAI1 peptide were 20 μ M and 160 μ M, respectively. *Lane 1*, Flag-RhoG input.

Reviewer #2 (Remarks to the Author):

Adhesion GPCRs are a novel GPCR family with very large extracellular and intracellular regions. Emerging studies link them to various cellular processes. While there is increasing attention on the structures and functions of the extracellular regions, the structures and interactions of the intracellular regions has lagged behind. Although G-protein coupling occurs via the cytoplasmic tails, the very large intracellular regions should be involved in other functions.

This manuscript provides a first structure/function analysis of the intracellular interactions of the important subfamily of BAI proteins. Weng et al. found an evolutionarily conserved fragment located in the intracellular region of BAI1 specifically binds to the RBD-ARR-ELMO (RAE) tandem of ELMO2 via biochemical characterization of BAI/ELMO2 interaction. ITC results show that BAI1-EBD binds to ELMO2 in a 1:1 molar ratio. The authors then solved the crystal structures of the RAE tandem of ELMO2 and its complex with conserved BAI1 intracellular region, respectively. The structures indicated that the RBD, ARR, ELMO domains of ELMO2 form a structural and functional supramodule required for BAI1 binding. Further structural analysis revealed the interfaces between ELMO2-RAE and BAI1-EBD. On a functional level, the authors used myoblast fusion assay to examine the functional relevance of the BAI/ELMO binding interfaces. The results show that mutants deficient in binding failed to

restore myoblast fusion. Finally, amino acid sequence analysis found that a GPR128 C-terminal fragment displaying high sequence similarity to BAI1-EBD, and the GPR128/ELMO2 complex also employs the binding mode uncovered by the BAI1/ELMO2 complex structure.

Whether this interaction can be generalized for all adhesion GPCRs or not, this study will be a first to attract the needed attention to the unknown functions of the intracellular regions of adhesion GPCRTs. The structural and functional work is decent and I recommend publication after the below points are addressed.

We thank the reviewer for the warm and encouraging comments on our work.

Abstract:

1) “The adhesion G protein-coupled receptors (aGPCRs) play crucial roles in diverse cellular processes through the G protein-independent signaling pathways, though the underlying molecular mechanism is not well understood.”

Please revise. Not all G-protein independent pathways are TM dependent. There are TM independent functions of adhesion GPCRs which are unrelated to the current study. Please be specific and cite such functions in the intro.

*We agree with the reviewer that not all the TM independent cellular functions of aGPCRs are ELMO/DOCK dependent. We have specifically revised the **Abstract** as: “The brain-specific angiogenesis inhibitor (BAI) subfamily of adhesion G protein-coupled receptors (aGPCRs) plays crucial roles in diverse cellular processes including phagocytosis, myoblast fusion, and synaptic development through the ELMO/DOCK/Rac signaling pathway, though the underlying molecular mechanism is not well understood.”*

*We have further expanded the content on the PDZ-binding motif-mediated intracellular signaling pathways in the **Discussion** section.*

2) “..another aGPCR-GPR128 was identified as a new upstream receptor for the ELMO family scaffolds, most likely with a recognition mode similar to that of BAI/ELMO pairs, thus suggesting a general action mode for aGPCRs. “

The word general mechanism should be used carefully. 2 out of 33 adhesion GPCRs does not qualify for general.

We thank the reviewer for alerting us the overstatement in our manuscript. We have made proper changes to remove such overstatement in the revised manuscript.

Other:

3) Please use better grammar.

We have taken extra care to correct typos and grammatical mistakes in our revised manuscript.

4) A comprehensive sequence analysis of the intracellular regions of all adhesion GPCRs will be a great supplement for this paper.

This is a wonderful suggestion. As suggested, we have made a comprehensive sequence analysis of the intracellular regions of all adhesion GPCRs (33 members of 9 sub-families)(**FIG-III** below, revised **Supplemental Figure 8**). We found that many of the aGPCR members contain a PDZ-binding motif (PBM) in the very C-terminus of the cytosolic tails, indicating that these aGPCRs may exert effects on intracellular signaling pathways via PDZ-containing proteins. It is also noted that, several aGPCRs possess the conserved sequence motifs with unknown binding target(s). Further studies may be needed to investigate whether these conserved motifs are involved in the cellular functions of these aGPCRs. We have included these points in the revised manuscript.

5) Is the ELMO binding peptide at the very C-terminus? How many other residues until the end of the C-terminus. Some adhesion GPCRs have been proposed to have PDZ binding sites at the C-terminus. A general analysis will be great.

As shown in **FIG-III** above and **FIG-IV** below (revised **Figure 1a**), the ELMO-binding site is located at the C-terminal part of the intracellular region of BAI1, ~ 80 amino acid residues away from the typical PDZ-binding site of BAI1. As suggested, a general sequence analysis of the intracellular regions of all adhesion GPCRs has been included in the revised manuscript as **Supplemental Figure 8**.

FIG-IV (revised Fig. 1a). Domain organization of BAI1. RGD, Arg-Gly-Asp integrin-binding motif; TSR, thrombospondin type 1 repeat; HBD, hormone-binding domain; GAIN, GPCR autoproteolysis-inducing domain; EBD, ELMO-binding domain. In the drawing, the structure-based sequence alignment of BAI1-EBD from different species and orthologs is shown. The absolutely conserved and conserved residues are colored red and green, respectively. Residues involved in binding to ELMO2 are annotated below as blue dots. The residues with clear electron densities in the structure are highlighted with gray box.

6) Page 6: Typo (aa 1167-1502)

Noted with thanks, and the proper change has been made in the revised manuscript.

7) Use of the word coupling interfaces should be clarified. Binding will be a better word.

As suggested, we have made the corresponding change in the revised manuscript.

8) *The proper expression and folding of the BAI mutants and ELMO mutants is critical for the validity of the experiments that report a mutation abolishes binding. A mutation that unfolds the protein will also give the same result although it is not at the binding interface. Please provide expression/folding information.*

To address the reviewer's concern, we freshly purified the wild-type, E198A, L202D, L244D and H284A form of ELMO2-RAE proteins which all behaved well in solution. Further circular dichroism (CD) spectrum data demonstrated that the mutants of ELMO2-RAE did not induce significant conformational changes on the overall folding of ELMO2-RAE (FIG-V). We have included this point in the revised manuscript.

FIG-V. Mutants of ELMO2-RAE did not induce significant conformational changes of ELMO2-RAE. (A) SDS-PAGE of freshly purified wild-type and mutants of ELMO2-RAE. (B) circular dichroism (CD) spectrum of wild-type and mutants of ELMO2-RAE.

9) How many ELMOs are there and which ELMO is used? GPR128 looks like it binds to ELMO less. Can it bind to other ELMOs more?

Following the reviewer's suggestion, we freshly purified GST-tagged RAE tandem of ELMO1-3, and found that all the RAE tandems could effectively bound to GFP-GPR128-CT (FIG-VI below). We have included the data in the revised manuscript as **Supplemental Figure 7**.

FIG-VI (revised Fig. S7). RAE tandem of ELMO1-3 all bound effectively to cytosolic tail of GPR128. PD, pull down.

10) GPR128-CT in Fig 6. Is this full-length CT?

Yes. We have included the information in the revised manuscript.

11) Does BAI-EBD bind to G protein? If it does, how BAI intracellular region choose different binding partners and influence the signal transduction?

This is an excellent point! Although aGPCRs belong to the GPCR superfamily, only a few of them have been linked to G proteins. As to BAI-aGPCRs, BAI2 was found to activate nuclear factor of activated T cells (NFAT, a transcription factor involved in immune responses) via $G\alpha_{16}$ (Okajima, D. et al. 2010); BAI1 was found to activate Rho pathway via a $G\alpha_{12/13}$ -dependent mechanism (Stephenson, J.R. et al. 2013). However, there is no evidence showing that BAI-aGPCRs could directly bind to $G\alpha$ protein. At atomic level, several GPCR- $G\alpha$ complex structures have been solved including rhodopsin- G_i (Kang, Y. et al. 2018), β_2AR - G_s (Rasmussen, S. et al. 2011), μOR - G_i (Koehl, A. et al. 2018), and $A_{2A}R$ - G_s

(Carpenter, B. et al. 2016) complexes (**FIG-VII** below). In these structures, $G\alpha$ specifically contacted with the intracellular loops of 7TM region of GPCRs.

The ELMO-binding domain (EBD) of BAIs are nearly 300 residues away from the 7TM. Moreover, the 300-residue linker between 7TM and EBD is predicted to be the unstructural region. Therefore, even if BAIs could bind to $G\alpha$ directly, the EBD domain is most likely not involved in the BAI- $G\alpha$ crosstalk. Future investigations are required to dissect the relationship of the two pathways. However, in the present study, we mainly focus on the BAI/ELMO signaling and are not going to pursue it further. We hope the reviewer could accommodate it.

FIG-VII. Selected atomic structures of GPCR-G α complexes

References

Carpenter, B. et al. (2016) Structure of the adenosine A(2A) receptor bound to an engineered G protein. *Nature* 536(7614):104-7.

Koehl, A. et al. (2018) Structure of the μ -opioid receptor-Gi protein complex. *Nature* 558(7711):547-552.

Kang, Y. et al. (2018) Cryo-EM structure of human rhodopsin bound to an inhibitory G protein. *Nature* 558(7711):553-558.

Okajima, D. et al. (2010) Brain-specific angiogenesis inhibitor 2 (BAI2) may be activated by proteolytic processing. *J. Recept. Signal Transduct. Res.* 30, 143–153.

Rasmussen, S. et al. (2011) Crystal structure of the β 2 adrenergic receptor-Gs protein complex. *Nature* 477(7366):549-55.

Stephenson, J.R. et al. (2013) Brain-specific angiogenesis inhibitor-1 signaling, regulation, and enrichment in the postsynaptic density. *J. Biol. Chem.* 288, 22248–22256.

12) Why some BAI-EBD variants show as multiple bands on the gel, such as “H1485A” in figure. 4, panel e.

We agree with the reviewer that the purities of some proteins (e.g., GST-BAI1-H1485A) in the GST-pull down assay were not good enough. We have improved the assay to address the reviewer’s concern by showing that fresh purified wild-type GST-BAI1-EBD could effectively bind to ELMO2-RAE, while the mutants of BAI1 (i.e., R1489A, H1485A, I1483D, Y1475A) all abolished the interactions (**FIG-VIII**). This result is now included as **Fig.4e** in the revised manuscript.

FIG-VIII (revised **Fig. 4e**). GST-pull down-based assays showing that key residues involved in the ELMO2-RAE/BAI1-EBD interface are required for the intact interaction.

REVIEWERS' COMMENTS:

Reviewer #1 (Remarks to the Author):

This already outstanding paper has been further improved. I have no additional comments and congratulate the authors for a beautiful piece of work.